# A construction heuristic for the capacitated Steiner tree problem

**Simon Van den Eynde****, Pieter Audenaert\*, Didier Colle, Mario Pickavet**

IDLab, Ghent University - imec, Ghent, Belgium

\* pieter.audenaert@ugent.be

## Abstract

Many real-life problems boil down to a variant of the Minimum Steiner Tree Problem (STP). In telecommunications, Fiber-To-The-Home (FTTH) houses are clustered so they can be connected with fiber as cost-efficiently as possible. The cost calculation of a fiber installment can be formulated as a capacitated STP. Often, STP variants are solved with integer linear programs, which provide excellent solutions, though the running time costs increase quickly with graph size. Some geographical areas require graphs of over 20000 nodes—typically unattainable for integer linear programs. This paper presents an alternative approach. It extends the shortest path heuristic for the STP to a new heuristic that can construct solutions for the capacitated STP: the Capacitated Shortest Path Heuristic (CSPH). It is straightforward to implement, allowing many extensions. In experiments on realistic telecommunications datasets, CSPH finds solutions on average in time $O(|V|^2)$, quadratic in the number of nodes, making it possible to solve 50000 node graphs in under a minute.

## 1 Introduction

When building a greenfield point-to-point fiber network, the goal is to connect several buildings to a central cabinet. The major costs are the cost to open the road and build a pipe, the *trenching cost*, and the cost of the fiber, the *fiber cost*. In point-to-point networks, each building is connected to the central cabinet through a single fiber (there are no splitters). Once a road is trenched, multiple fiber bundles can go through the same connection without having to trench multiple times. This is similar to the Minimum Steiner Tree Problem (STP), see Problem 1, where each edge weight is counted at most once. On the other hand, each extra meter of fiber costs more money. In general, it is much easier to optimize fiber costs (e.g. shortest paths from the fiber cabinets to the buildings) compared to the trenching cost: solving a rooted STP with as root the central cabinet and with as terminals the buildings is NP-hard. To make sure that the proposed constraints are rooted in real-world problems, we worked together with Comsof, a telecommunications company that develops software to design fiber networks.

Due to the cable size, there are sometimes restrictions on the maximal number of fibers that run through an edge. This can also happen in non- or semi-greenfield installation, where there is still some room in existing cables but not much. We say that edges have a maximal fiber capacity.

**Data Availability Statement:** The Vienna dataset is available at https://homepage.univie.ac.at/ivana.ljubic/research/STP/ The LIN dataset is available at the Steiner Tree library at http://steinlib.zib.de/steinlib.php?LIN The PUC dataset is available at the

Steiner Tree library at http://steinlib.zib.de/steinlib. php?PUC.

**Funding:** This research was partly funded by the Ghent University IOP project "Modelling Uncertainty in Hub Location Planning through Interdisciplinary Research", by the VLAIO project "Comsof Autonomous Planning Agent" and by the UGent-project BOF/STA/202009/039. The funders had no role in study design, data collection and analysis, decision to publish, or preparation of the manuscript.

**Competing interests:** The authors have declared that no competing interests exist.

This capacitated variant of the STP is described below as Problem 2. This specific variant is also used for wind-farm cabling as is shown by [1], where a transformation from STP to an integer linear program (ILP) formulation is proposed. Both the wind farm and the telecommunications examples use sparse graphs, where complexity-wise the number of vertices $|V|$ equals the number of edges $|E|$, in big-O notation, this is written as $O(|V|) = O(|E|)$. Therefore, we will restrict our tests to sparse graphs, and in particular, we will focus on the telecommunications example by mainly considering road networks for testing graphs. The objective of this paper is to find a heuristic that can solve capacitated STP quickly. It does so by extending the shortest path heuristic for STP to incorporate capacities. The result is a new heuristic, the Capacitated Shortest Path Heuristic (CSPH). Furthermore, benchmark data for the STP is adapted by adding capacities, so it can be used to measure the quality and time complexity of CSPH.

**Problem 1 (Steiner Tree Problem (STP))** *Consider an undirected simple connected graph G = (V, E), a cost function $c(e) : E \rightarrow \mathbb{R}_{\geq 0}$ on the edges and a set of terminals $T \subseteq V$. The minimum Steiner tree problem in graphs then requires finding a minimal-cost tree that connects all terminals.*

In this paper, it can also be called the Steiner problem, the Steiner tree problem, or STP. The minimum Steiner tree problem is a classical NP-hard problem, see [2].

**Problem 2 (Capacitated Steiner Tree Problem (CSTP))** *Consider an undirected simple connected graph G = (V, E), a cost function $c(e) : E \rightarrow \mathbb{R}_{\geq 0}$ on the edges, a capacity function $cap(e) : E \rightarrow \mathbb{N}_{>0}$, a root $r \in V$ and a set of terminals $T \subseteq V$ with demands $T \rightarrow \mathbb{N}$. We say that edge e has capacity $cap(e)$. The capacity of an edge e restricts the maximum number of root-terminal paths (with as cost the terminal demand) over this edge to $cap(e)$. The capacitated minimum Steiner tree problem in graphs requires finding a minimal-cost tree that connects the terminals such that the edge capacities are respected.*

The next section begins by discussing the related work: we discuss why we took a heuristic approach and which papers most inspired our heuristic. Section 3, describes the Shortest Path Heuristic, a well-performing heuristic for the STP. Section 4 details the main contribution of this paper the capacitated shortest path heuristic (CSPH), a fast heuristic for solving the CSTP. The following section contains several possible extensions to incorporate additional constraints in the code and demonstrate the flexibility of CSPH. Next, in Section 6 the experiments and datasets are described. The paper uses open-source datasets commonly used to benchmark STP problems and generates CSTP problems by adding fiber costs and capacities. The results of the experiments section are shown in Section 7. While it is difficult to estimate the quality of the solution, a rough estimate is provided. Furthermore, the section contains a comprehensive time analysis, theoretical as well as empirical. Lastly, in the conclusions, we summarize the problem, our solution, CSPH, and how it performs.

## 2 Related work

### 2.1 Steiner tree construction heuristics

A special case of the Steiner tree problem is the Minimum Spanning Tree problem (MST). MST can be solved optimally by the $O(|V| * \log(|V|))$ Prim algorithm with a binary heap. There also exists an average-case linear time $O(|E|)$ algorithm by [3], but it builds upon a rather convoluted MST verification algorithm. Several attempts have been made to create fast Steiner tree constructions. For example, for rectilinear graphs, a Steiner tree can be built in linear time, given an MST that fulfills certain shape properties, see [4].

A famous and well-performing Steiner tree construction heuristic is the shortest path heuristic (SPH) first proposed in 1980 by [5]. An improved version was published in 2002 in [6] under the name Prim-Improved. It is this version that is presented in Algorithm 1. The reason

that we specifically chose SPH is twofold. First, in [6], a that proposes and compares several fast STP heuristics, SPH is the best performing algorithm for VLSI-type graphs. Second, in 2013 the 11th DIMAC challenge on Steiner trees was launched. Several interesting papers on STPs have been developed thanks to this challenge. [7] presents SCIP-Jack, a mathematical program solver that can tackle several different Steiner tree problems. Similarly, the versatile iterative heuristic presented by Pajor, Uchoa, and Werneck in [8] performs well on several Steiner tree problem classes by building several trees, performing local updates, and combining the best trees efficiently. As the initial tree-constructing heuristic, they suggest using the shortest-path heuristic (SPH).

## 2.2 Alternative Steiner tree approaches

In recent years, a lot of deep learning models have attempted to solve hard combinatorial graph problems. Often by building deep neural nets that take graphs as input, commonly labeled as graph convolutional networks. Unfortunately, we do not think that deep neural nets are the right approach for this large-scale combinatorial problem. To show why we will detail the work of several papers on learning heuristics for combinatorial optimization problems and why their models are not sufficient to find decent solutions to CSTP with more than 5000 graph nodes.

The authors of [9] claim that it is currently not possible to scale end-to-end deep learning to real-world instances. So, they propose to train a deep learning model on small problem instances to then generalize these in a zero-shot fashion. The paper details in which ways this can be done and which parts require more attention. Since this is an exploratory paper on this novel way of approaching combinatorial problems with deep learning, the results are not that good compared to classical approaches. In particular, their optimality gap for the TSP is at 10% for graphs with 200 nodes. To summarize: the presented approach may hold the future for solving large-scale combinatorial problems, but this is not yet feasible. Moreover, models for these kinds of graphs are mostly not size-generalizable: if the model was built on small training graphs, it will underperform on larger graphs. This paper aims for graph sizes up to 50, 000 nodes. The current graph models only test for sizes up to 500 nodes. Case in point, in 2019 the paper [10] provides new best results by combining a neural net with attention layers and a custom learning approach. They get very good results but only look at graphs up to 100 nodes.

Two papers published in 2021 attempt to solve the Steiner tree problem directly. The first paper considers several black-box neural nets (feed-forward, graph neural, graph convolutional, and graph attention), which have a significantly worse approximation ratio than the 2-approx [11]. The second type of model the paper considers is an extension of the 2-approx method where additional nodes are generated by neural nets. While this method has a good approximation ratio, it requires executing multiple 2-approx iterations, significantly impacting the running time. The second paper approaches the problem differently by embedding graphs into a low-dimensional vector and feeding them to a reinforcement learning model [12]. Their model gets good approximation results on several small graphs, but it seems to be about 1.5 times slower than 2-approx. Furthermore, the authors show that the model is capable of some generalization, however, the paper only experimented on graphs with up to 150 nodes.

## 2.3 Capacitated Steiner tree problem

The capacitated Steiner tree problem is hard and at the same time rather specific, making it a less popular subject. From the reduction to STP, CSTP is NP-complete. Contrary to STP, for which [13] has shown that it can be approximated up to a factor of 1.39 with an LP

approximation model, [14] has proven the inapproximability of CSTP. While CSTP can be reduced to STP, it can also be reduced to the capacitated minimum spanning tree (CMST) problem. An overview of CMST heuristics was written by [15].

A more general variant of CSTP is much more famous: the capacitated fixed-charge network design problem (CFNPD) where instead of a single root and a set of destinations, there are origin-destination pairs. Setting all origins to a single root reduces the problem to CSTP. Because of the large number of constraints, CFNPD is usually solved with ILPs, such as by [16]. The disadvantage of ILPs is their complexity, for example in [17], where they use a combination of mathematical programming techniques and heuristic to find provably high-quality solutions to the fixed charge network flow problem, calculating a near-optimal solution for a 500-node 2500-edge graph easily takes 15 minutes or more. Similarly, [18] takes about 10 minutes to solve problem instances with 100 nodes and 400 edges when using their MIP solver with a novel local branching metaheuristic.

## 3 Shortest Path Heuristic

In this section, a heuristic to solve the Steiner tree Problem will be discussed. Several path heuristics were considered, such as the zoom-in approach and the iterative pathfinding [19]. For this paper, SPH was chosen over other heuristics because [6] concluded that it (under the name of prim-improved) was their best pick after comparing several fast STP construction heuristics—including the distance network heuristic and adaptations of MST algorithms Prim, Kruskal, and Boruvka. Moreover, [8] implemented SPH in a competitive Steiner Problem solver. The main drawback of SPH is that tests from [6] have indicated that the performance of SPH can decrease for some harder instances. However, the instances under consideration will be road networks, which are considered easy instances: the graphs are generally sparse and have low max and average degrees.

SPH builds a solution tree terminal by terminal, see Algorithm 1 for the pseudocode. Starting from a partial solution tree *PST*, SPH searches for the terminal closest to *PST* with a Dijkstra-variant (lines 7-13). Once the closest terminal is found, the terminal and its path to the *PST* are added to this tree. As [6] has demonstrated, it is not necessary to fully restart the Dijkstra search each time a new path is added to the *PST* because the upper bounds determined by the Dijkstra search do not change when a path is added to the *PST*. So, after setting the upper bound of the new *PST*-nodes to 0 and putting these nodes once again in the Dijkstra heap, the search can continue (lines 14-20).

**Algorithm 1**: Shortest Path Heuristic

```
 input: A network G, a root root and a set of terminals T
 output: A solution tree containing all terminals
1 foreach node ∈ G do
2   SteinerUpperBound (node) ← inf;
3 SteinerUpperBound (root) ← 0;
4 node ← root;
5 Initialise heap;
6 while T not empty do
7   while node ∉ T do
8     foreach neighbour neigh of node do
9       newDist ← EdgeCost (node, neigh) + SteinerUpperBound (node);
10       if newDist < SteinerUpperBound (neigh) then
11         SteinerUpperBound (neigh) ← newDist;
12         Add (newDist, neigh) to heap;
13     (_, node) ← Pop heap;
14   // ADAPT-phase, at this point node is a terminal
15   Remove node from T;
```

```
16   path the path from node to the solutionTree;
17   Add path to the solutionTree;
18   foreach node ∈ path do
19     SteinerUpperBound (node) ← 0;
20     Add (0, node) to heap;
21 Return solutionTree;
```

For a fast version, it is possible to use the following data structures. To enable the Dijkstra search, SPH uses a binary *heap* with distance-node pairs, sorting on the distances. The same node can be added twice with the same distance, which can be corrected by the node-distance map *SteinerUpperBound*. When a node is polled from the heap, the distance can be verified with the node's distance in the map and passed over if unequal. As can be derived from the name, the map will always contain an upper bound for the distance to the partial solution tree. Lastly, a Dijkstra search tree (or some kind of predecessor map) must be maintained to retrieve the root-terminal path once a terminal is polled.

In every iteration, SPH finds the terminal closest to the tree that is already built. In worst case, this heuristic runs in time $O(|T|^*(|E| + |V| * \log(|V|)))$. However, empirically—on several graphs, including road networks—it runs independent of $T$, in time $O(|E| + |V| * \log(|V|))$, see [6].

## 4 Capacitated Shortest Path Heuristic

The Capacitated Steiner Tree Problem (CSTP) definition can be found in Section 1 at problem 2. The difference with STP is threefold. Terminals now have demands, edges have capacities, and the fiber cost is considered in the optimization objective. Each edge $e$ has an associated capacity $cap(e)$. Then, for a solution to be valid, for each edge $e$, the sum of the demands of the terminals it connects to the root (i.e. the flow) is at most $cap(e)$. We say that, in a partial solution tree, the terminals incur a *flow* on the edges. The *leftover capacity* of an edge is defined as its capacity minus the current flow on the edge.

This section will derive an adaptation of SPH for CSTP. This heuristic, CSPH, calculates a quick and valid solution with an acceptable error rate. As seen in Section 3, SPH builds a Steiner tree solution by connecting terminal after terminal to the tree. Each new terminal is found by an adapted Dijkstra starting from the partial solution tree. CSPH, see Algorithm 2, maintains this idea and builds a solution tree terminal by terminal. The next terminal to add is found by performing a Dijkstra search (with adapted edge costs) from the partial solution tree. Once the cheapest path to a terminal is selected, it is verified whether this path has sufficient capacity for the terminal demand. If so, the terminal is added to the solution tree and the next terminal will be searched for. If not, some of the edges will be labeled as full (they are effectively removed from the graph for the remainder of the search) and the search is repeated. Once all terminals have been connected, the solution tree is returned.

**Algorithm 2**: Capacitated Shortest Path Heuristic

```
input: A network G, a root root and a set of terminals T
output: A solution tree containing all terminals
1 foreach node ∈ G do
2   SteinerUpperBound (node) ← inf;
3 SteinerUpperBound (root) ← 0;
4 node ← root;
5 while T not empty AND heap not empty do
6   while node ∉ T do
7     foreach neighbour neigh of node do
8       if neigh ∈ solutionTree and Edge(node,neigh) ∉ solutionTree then
9         continue;
```

```
10        newDist ← EdgeCost (node, neigh) + SteinerUpperBound (node);
11       if newDist < SteinerUpperBound (neigh) then
12         SteinerUpperBound (neigh) ← newDist;
13         Add (newDist, neigh) to heap;
14     (_, node) ← Pop heap;
15   path ← the path from node to the root;
16   if Each edge on path has sufficient leftover capacity (must be at
least 1) then
17     // ADAPT-phase
18   else
19     // RESET-phase
20 Return solutionTree;
```

Instead of just a single tree, two trees are maintained. The Dijkstra tree is used during the search and can change the structure in the ADAPT and RESET phases. The SolutionTree, on the other hand, maintains the partial solution and can only increase in size (once added, nodes or edges will not be removed). During the Dijkstra search, some nodes are skipped over to prevent the creation of loops and assure the solution is a tree, see line 8. Furthermore, while in SPH, when a new terminal is found with the Dijkstra search, the terminal is immediately added to the solution (ADAPT phase, line 17)), in CSPH, first the leftover capacities of the edges on the root-terminal path have to be validated.

Only if each edge has sufficient leftover capacity can the path be added to the solution with an extra step to record the capacities (ADAPT-phase, line 17). However, if an edge $e_{full}$ has insufficient leftover capacity, it is removed from the graph (RESET-phase, line 19), though if it is already present in the solution tree, it will remain a part of the solution. Moreover, all descendant edges in the Dijkstra search tree are removed from the tree, their upper bounds are set to infinity, and all unaffected neighbors (in the network) of the removed nodes are added to the heap with their current distance. The descendant edges of $e_{full}$ are the edges $e_d$ such that the path from a node incident to $e_d$ to the root passes through $e_{full}$. Note that by removing edges due to capacity constraints, the heuristic is not guaranteed to find a solution.

Now we will investigate the edge cost function. The total edge cost is the trench cost plus the fiber cost. The trench cost of an edge has to be paid only once—independent of the number of connected terminals. In other words, when an edge is added to the solution tree, its trench cost is reduced to 0. If the solution tree is represented as a predecessor map, checking whether an edge is present in the solution tree can be done in constant time. The fiber cost is based on the terminal demand, we use the average demand as an estimation.

## 5 Extensions and improvements

### 5.1 Cost requirements

While sometimes capacity requirements are strict, it frequently occurs that overflow just creates an additional cost. This cost is generally quite high, for example in telecommunications, a capacity overflow would indicate that a street has to be broken up for a second time, requiring expensive man-hours as well as burdening the users of the street. Still, it might be the best solution at hand. Such a penalty can be implemented as follows. During the RESET phase, when it is discovered that an edge $e$ has insufficient leftover capacity for a terminal demand, $e$ is labeled *full* instead of removing it. The Dijkstra search tree is still updated (reset), just like the upper bounds and the heap. Then, a dedicated edge cost method can take into account whether an edge is present in the solution tree and whether it is full. If full, a custom penalty can be added. Because the search tree was partially reset, it is possible to change the edge cost to any positive value without breaking the heuristic. Such a change guarantees that the heuristic always finds a solution.

Another change to the cost function can be implemented by not setting the cost to 0 for edges already present in the solution tree. This is necessary if re-using this edge for a new root-terminal connection incurs a small cost. This can be resolved in the same cost-function as described above, and by taking this into account when adding a path to the solution tree. However, when using a value for the re-use cost that is higher than the single-use cost, the costs will not be taken into account correctly, because the heap prioritizes small distances and no update is forced, contrary to when an update is forced in the RESET-phase each time capacity over-flow increases the cost. So, as common in Steiner problems, keep the re-use cost smaller than the single-use cost.

## 5.2 Capacity—Demand structure

Several possible extensions can easily be tackled by this heuristic. First, the terminal demands could be any integer. The CSPH heuristic can be adapted to this purpose with some minor changes. In the ADAPT phase, when a path is added to the solution, the edge demands have to be updated according to the terminal demand. Moreover, in the RESET phase, all edges that have insufficient capacity for the current terminal are removed. This ensures that the algorithm chooses another path in the next iteration. The disadvantage is that some edges might be labeled *full* prematurely because of a terminal with large demand, while such an edge could still serve a terminal with a small demand.

The second implicit assumption CSPH relies on is that there exists a total order on the capacities, and in particular, that capacities are comparable. This assumption can be avoided by relying on edge demands instead of leftover capacities. Where the *edge demand* of an edge $e$ is defined as the combination of the terminal demands of those terminals for which the path from the root to the terminal contains $e$. When demands are integers, it holds that the leftover capacity of $e$ plus the demand on $e$ equals the $e$'s capacity. So for an edge $e$, instead of comparing $e$'s leftover capacity with $cap(e)$, the edge demand is compared directly with the original $cap(e)$. The two techniques differ because when using the leftover capacity, two different capacities have to be compared. When relying on edge demands, this is no longer the case. Practically, this makes it possible to have a different discrete capacity function for each of the edges. For example, CSPH could now theoretically handle a case where terminal demands can be of two different types, such as *red* and *blue*. Then, some edges can allow both red and blue, while some allow only blue and some only red. The heuristic will provide a valid solution to the problem, however, it is quite possible that in such cases the solution quality is not as high. Since a root-terminal path causes a reset if an edge cannot fit the terminal demand, there will be a reset each time red-only edges are found on the path for a blue terminal and vice versa. This effect worsens the more discrete types there are.

## 6 Experiments

### 6.1 Data collection

All three heuristics were tested on several datasets. The authors had no access to any identifying information on individuals. Since the aim is to swiftly build an approximate Steiner tree, large example graphs are preferred. The most relevant online dataset was the Vienna dataset by [20], available at the link: https://homepage.univie.ac.at/ivana.ljubic/research/STP/. This dataset consists of large real-world examples, for which optimal solutions are available in their paper. The Vienna dataset consists of the GEO instances and I instances, with respectively 23 and 85 problem instances. Both contain telecommunication deployment and infrastructure data from various Austrian cities, but the I instances also contain more rural areas with sparser infrastructure. Due to privacy reasons, the I instances of the Vienna dataset do not have any

**Table 1. Dataset descriptions; all numbers are averages over all the graphs in a dataset.** #*x* represents the number of elements in *x*.

| dataset | #graphs | #nodes | #edges | #edges/#nodes | Location info |
|---------|---------|--------|--------|---------------|---------------|
| lin | 37 | 8587 | 15956 | 1.78 | Y |
| puc | 50 | 1269 | 8694 | 6.35 | N |
| Vienna G | 22 | 25872 | 40581 | 1.55 | Y |
| Vienna I | 15 | 10409 | 15480 | 1.48 | N |

location data and thus have no distance information. Of the I instances, the 15 first instances are used in the experiments.

Our results are also tested on some VLSI instances found in the Steiner library. The dataset used, lin, was created by A. Lin. The lin dataset contains large rectilinear graphs. The graphs in this set are structurally similar, for example, they have almost the same average degree. A second Steiner Library dataset that is used in the experiments is the puc dataset. It is included to have some comparison with less sparse graphs.

Table 1 shows the average values of some common measures for all the datasets. The average number of nodes divided by the number of edges indicates that the average degree for all datasets, except puc, lies between 2.8 and 3.8, which is due to the inherent sparse nature of the graphs and the limited amount of preprocessing (only the Vienna dataset was partly preprocessed).

## 6.2 Experiment descriptions

All algorithms were implemented using Python 3.8 and executed in a single core on a laptop with a 2.9GHz CPU and 8GB RAM.

The experiments are controlled for four parameters: the graph, the number of terminals (as a percentage of nodes), the base capacity (as a percentage of terminals), and the capacity structure. The experiment starts with the selection of a graph, with its associated trenching and fiber costs. Then, the number of terminals is determined based on a percentage, selected from {1%, 6%, 11%, 16%, 21%}, of the number of nodes. The terminals are randomly chosen amongst the nodes, with unit demands. Next, each edge is assigned a capacity. First the base capacity $cap_b$ is decided as $\frac{|T|}{10}$ multiplied with 1, 2 or 3. Second, a capacity structure is chosen. When the structure is random, the capacity on each edge is a random number between 1 and $cap_b \times 3$. When the structure is leveled, the capacity on each edge is $cap_b \times level$ with *level* randomly chosen from 1, 2, 3, 4, 5. When the capacities are leveled, there are only 5 different capacities over the entire graph. This is often the case in telecommunications since cable and duct sizes have fixed capacities.

For each graph, CSPH is executed for 5 different terminal settings and a total of 6 capacity settings (two levels and three base capacities). Then, each setting is repeated 10 times with a different random seed (so capacities and terminals are assigned differently). On the same data, SPH and Dijkstra are executed, but since they cannot take the capacities into account, this step is skipped, resulting in a total of 50 runs for each graph.

## 7 Results

This section shows and discusses the results of the experiments in section 6. All figures, except for Fig 1, were generated with the Python package Matplotlib.

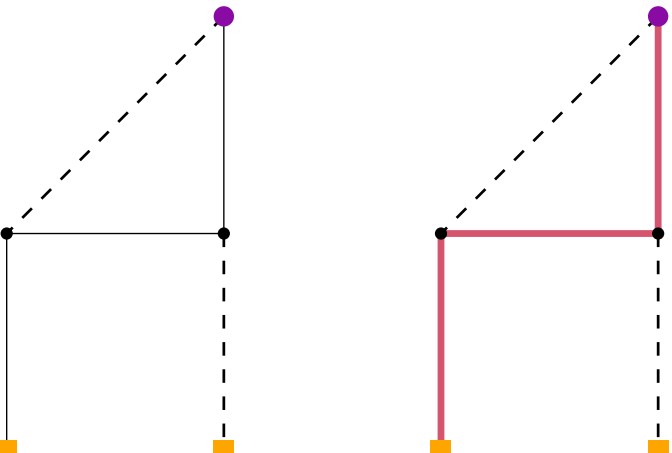

**Fig 1. CSPH example.** The large purple circle is the root. The yellow square nodes are the terminals. The left image has no connections. In the right image, a single path connects the root to a terminal.

### 7.1 CSPH quality performance

Since CSPH is a fast heuristic and the CSTP is NP-Hard, it is not possible to guarantee that a solution can be found, even if there is one available. A small example graph where you can see this is given in Fig 1. In this example, all edges have a capacity of one. The dashed edges have cost 10, while the other edges have cost 1. The left figure contains the original graph and the right figure contains the path that CSPH will find. This path connects a single terminal and blocks the path to the second terminal.

Since the experiments do contain capacities, CSPH fails to connect all terminals there as well. This can affect the runtime and the total cost. To not let this influence the time and cost averages, these failed instances are removed and the averages are, per graph and terminal settings, calculated separately. Otherwise, the settings with more successful CSPH executions would have a larger influence on the average.

The CSPH results are not compared with optimal solutions, since no algorithm was found that can generate these for large graphs. Instead, the cost subdivision (fiber and trenching) of the solutions is investigated and compared with the Dijkstra and SPH results. In particular, Dijkstra gives an optimal fiber cost, and [6] has shown that the optimality gap of the trenching cost of SPH on sparse graphs (VLSI-like) is generally less than 5%. However, both Dijkstra and SPH ignore capacity constraints, so their respective fiber and trench cost are possibly an underestimation of the actual optimal cost.

Fig 2 contains the average cost for the three methods on the Vienna G dataset, subdivided into fiber and trenching costs. The figure shows, as expected, that of the three methods, Dijkstra has the lowest trenching cost and SPH has the lowest fiber cost. The total cost of CSPH is lower than these of Dijkstra and SPH. Not only that, but, at least for this example, CSPH cost averages seem close to the non-capacitated average from Dijkstra and SPH, indicating that the solutions it generates are of good quality. Fig 3 shows the averages for the Vienna I dataset. In this case, the differences are less pronouncend and SPH scores as good as CSPH. Lastly, Fig 4, plots the averages for the puc dataset. The plot mirrors Fig 2 and the same conclusion can be drawn.

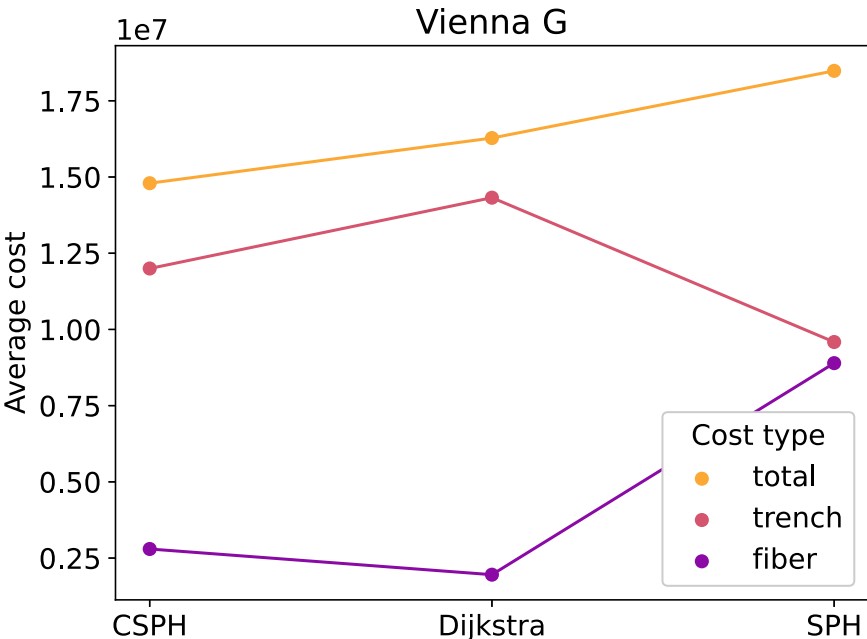

**Fig 2. Vienna G cost type analysis.** Different cost types (fiber, trench and total) for the CSPH, Dijkstra and SPH algorithms.

All in all, we think that these plots indicate that, if CSPH finds a solution, it is of good quality. In the telecommunications problem setting, often only a small subset of edges have a capacity restriction. This would make it much more likely that there exist many solutions, so CSPH is expected to perform well on these types of graphs.

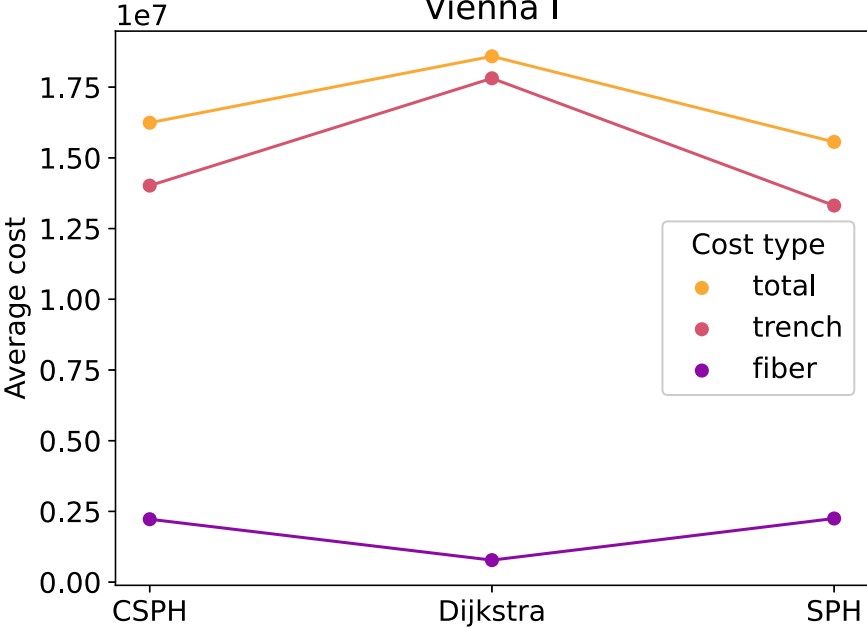

**Fig 3. Vienna I cost type analysis.** Different cost types (fiber, trench and total) for the CSPH, Dijkstra and SPH algorithms.

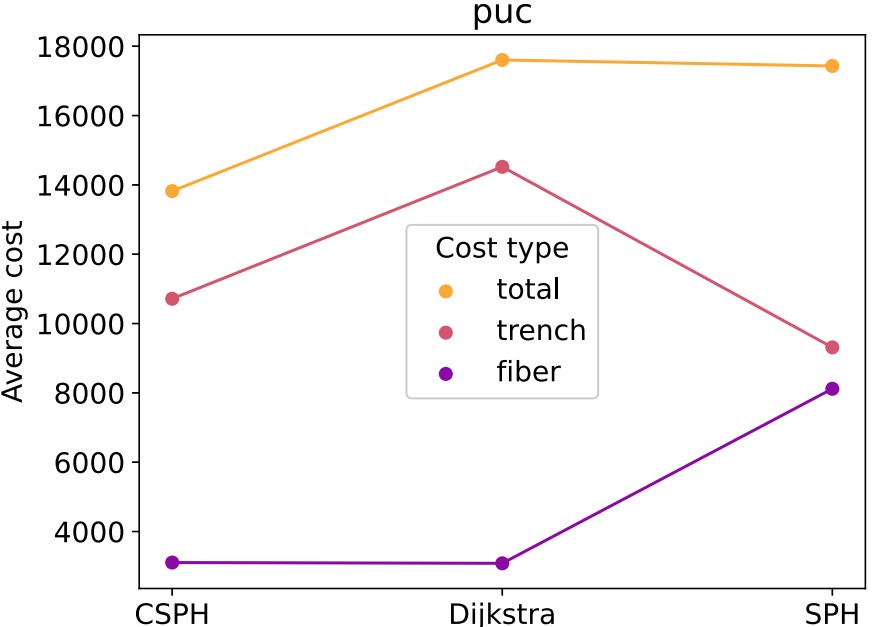

**Fig 4. Puc cost type analysis.** Different cost types (fiber, trench and total) for the CSPH, Dijkstra and SPH algorithms.

## 7.2 CSPH time analysis

Another important factor to assess the quality of an algorithm is its time complexity. How long does it take to find a solution? In this section, first, a theoretical worst-case analysis is

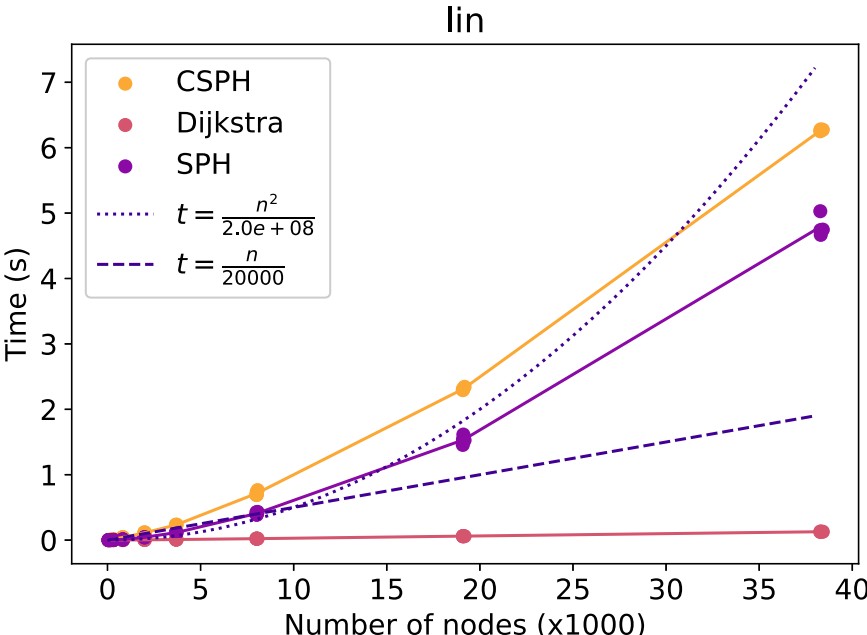

**Fig 5. Lin time evolution by method.** Evolution of the time cost of Dijkstra, SPH and CSPH. Includes a quadratic and linear function to compare trends.

conducted. Next, some computational experiments will demonstrate the effective time complexity of CSPH.

While the most interesting time analysis would be an average-case analysis, this seems to be a non-trivial task due to the large number of variables to consider. However, we can do a worst-case analysis. In the CSPH-algorithm (Algorithm 2), for the search for a single terminal, a Dijkstra heap implementation is used, with time complexity factor $O(|V| \times \log|V|)$. Now, the search for a terminal can fail if an edge reaches its capacity limit. After such a RESET phase, part of the search tree is removed and the Dijkstra search is repeated. This can happen at most once for every edge since after a RESET, the edge is labeled full and no longer considered. Lastly, each terminal is connected exactly once. This gives a total worst-case time complexity of $O(|T| \times |E| \times |V| \times \log|V|)$. This is equivalent with $O(|T| \times |V|^2 \times \log(|V|))$, since for sparse graphs $O(|E|) = O(|V|)$. In the experiments, $|T| = x^* |V|$ with $x$ the terminal ratio, so we would expect $O(|V|^3 \times \log(|V|))$ worst-case time complexity.

First, the general running time trend in relation to the number of nodes will be evaluated. The next figures show how the time costs of the methods evolve and compares them with a quadratic and linear function in the number of nodes. Each point on the figures is the average over all capacity, terminal, and seed settings for a particular graph and method. In Fig 5 the lin dataset running time is evaluated. The SPH running time takes here almost as long as the CSPH running time, indicating that CSPH most likely had very few resets. Moreover, the CSPH trend is subquadratic, though more time-consuming than a linear method would be. The same trends can be observed in Fig 6 on the Vienna G dataset, though the gap between CSPH and SPH is larger here. Lastly, in Fig 7 on the Vienna I dataset a different trend is visualized. Specifically, CSPH follows a roughly quadratic trend but remains far away from the worst-case $O(|V|^3 * \log(|V|))$ scenario.

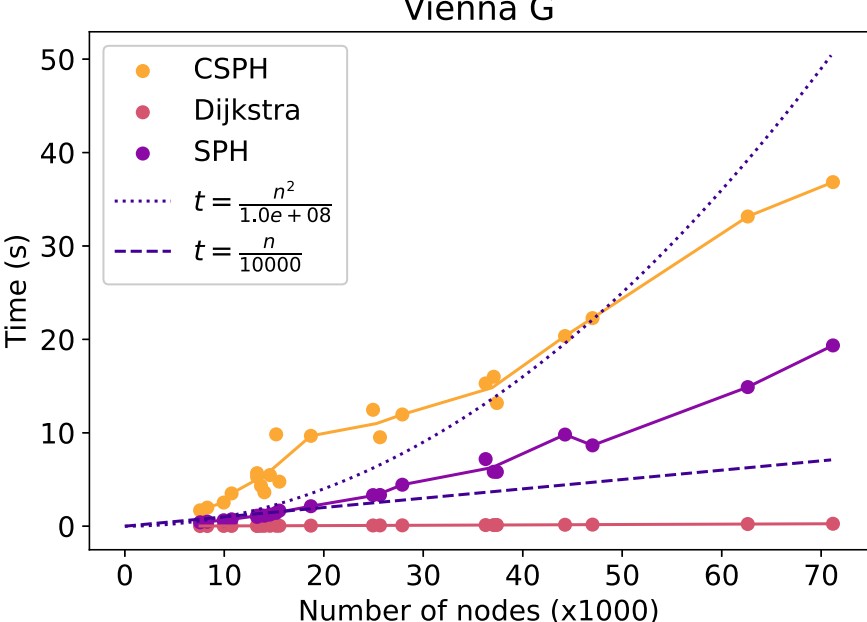

**Fig 6. Vienna G time evolution by method.** Evolution of the time cost of Dijkstra, SPH and CSPH. Includes a quadratic and linear function to compare trends.

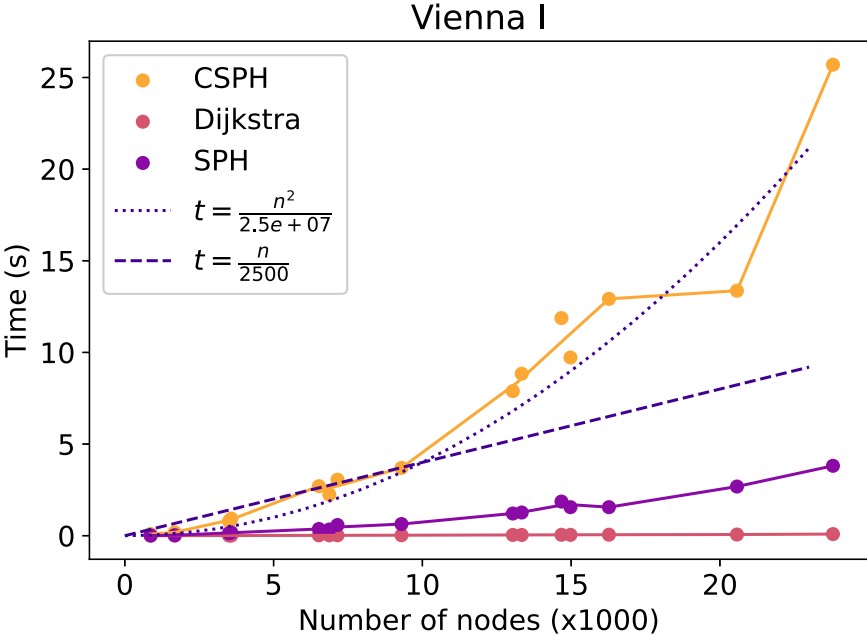

**Fig 7. Vienna I time evolution by method.** Evolution of the time cost of Dijkstra, SPH and CSPH. Includes a quadratic and linear function to compare trends.

Let us now take a more detailed look at how the number of terminals influences the running time. Fig 8 shows for the Vienna G dataset the average running time ordered by graph sizes and separated by the percentage of terminals. Furthermore, the number of terminals does not seem to have a strong influence on the running time. The running time difference between 1% terminals and 21% terminals for 30000 and 70000 nodes is about constant at 10s. The same plot for the Vienna I instances, see Fig 9, shows a less optimistic trend. In particular, the gap between the terminal ratio lines increases when the number of nodes increases. This shows that for the Vienna I dataset, the number of terminals has a significant impact on the running time.

The final figures show how the capacity structure and total capacity influence the running time. Fig 10 shows the running time of CSPH and the number of resets it took on average to find a solution for the G307 graph of the Vienna dataset with 7830 terminals. It is remarkable how much difference there is in time between the leveled approach, where there are only 5 possible capacities versus the random approach where practically every edge has a different capacity. The main reason why this reduces resets is that when CSPH resets a path, it labels all edges with no leftover capacity as full, and this will affect more edges in the structured approach. A similar pattern emerges in Fig 11 where graph I003 of the Vienna I instances is evaluated. However, the difference in running time is less pronounced. The resets are scaled differently, so there is still a large difference in resets. In both figures the observed standard deviations show that the timing results are sufficiently robust and that, for example, the patterns described above still hold when taking into account the deviations.

## 8 Conclusions

The Capacitated Steiner Tree problem (CSTP) extends the Steiner tree problem by limiting the capacity of each edge, adding demands to terminals, and including shortest path costs. One

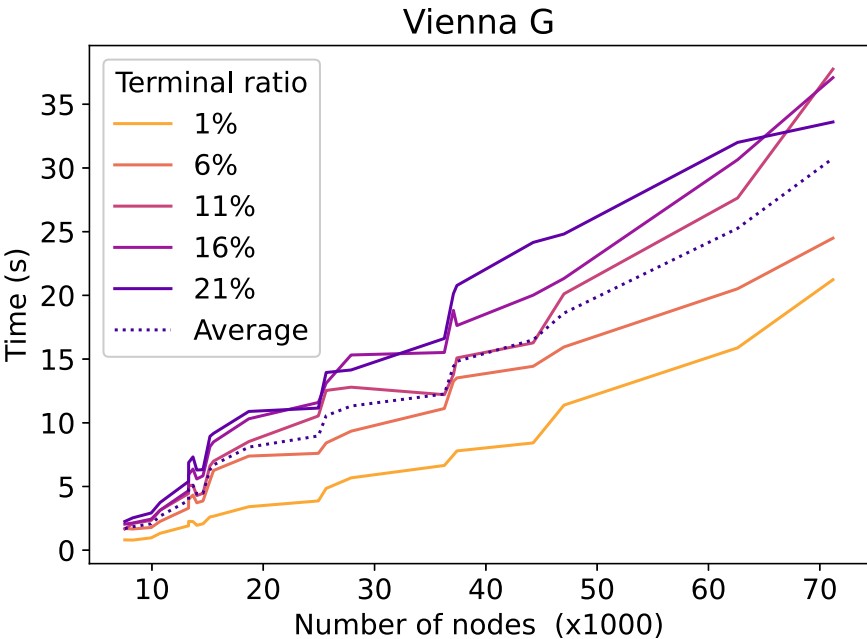

**Fig 8. Vienna G CSPH time evolution by terminal ratio.** The time in relation to the number of nodes. For each line, a different percentage of nodes is a terminal.

practical application of this problem is the optimal placement of Fiber-To-The-Home (FTTH) cables. There is a fixed cost per edge to trench the roads and install cables, a length-based cost for the fiber materials, and a capacity for the number of fibers that can pass through a cable. CSTP is proven to be inapproximable, so this paper focuses on the realistic FTTH variant,

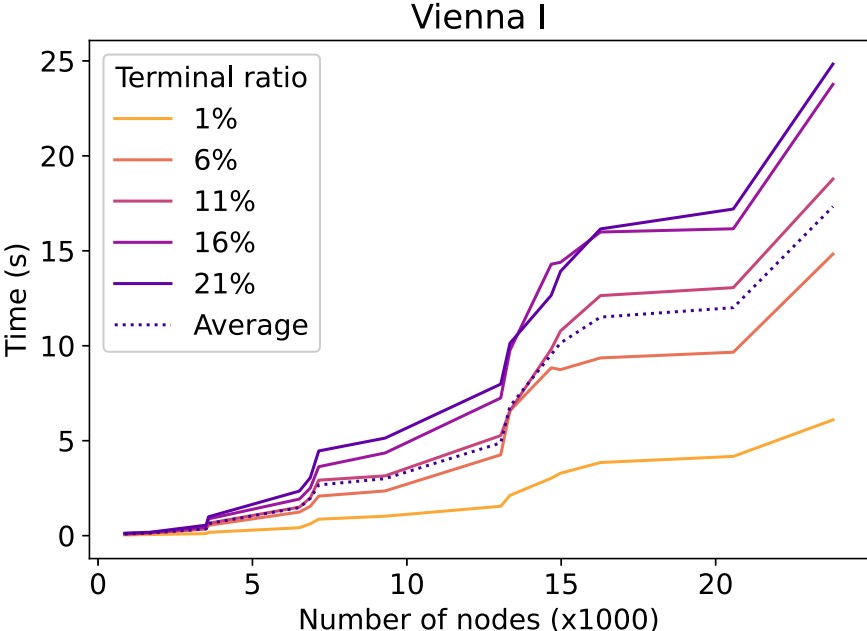

**Fig 9. Vienna I CSPH time evolution by terminal ratio.** The time in relation to the number of nodes. For each line, a different percentage of nodes is a terminal.

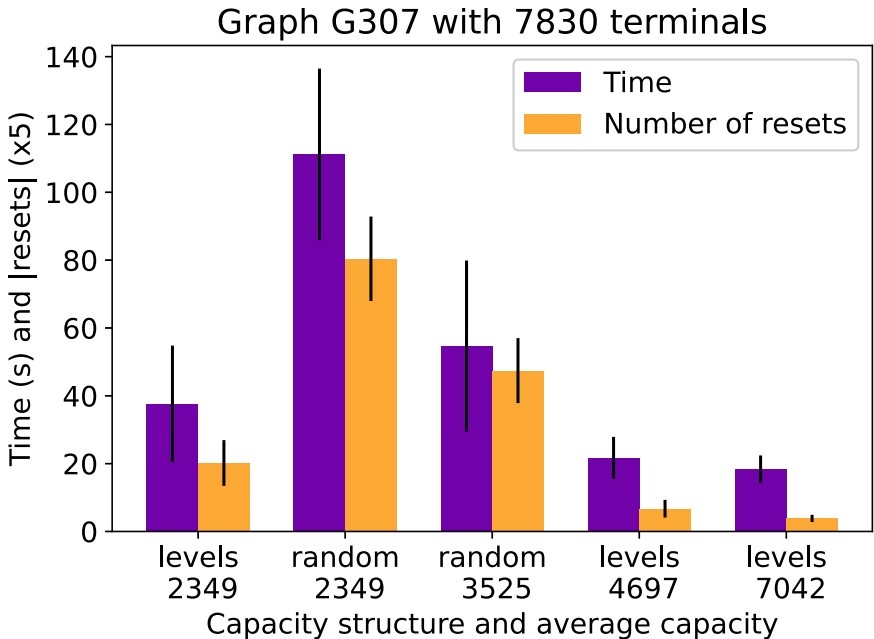

**Fig 10. Vienna G307 time compared by capacity.** The time (purple bars) and number of resets (yellow bars) for 5 different capacity settings for the G307 graph.

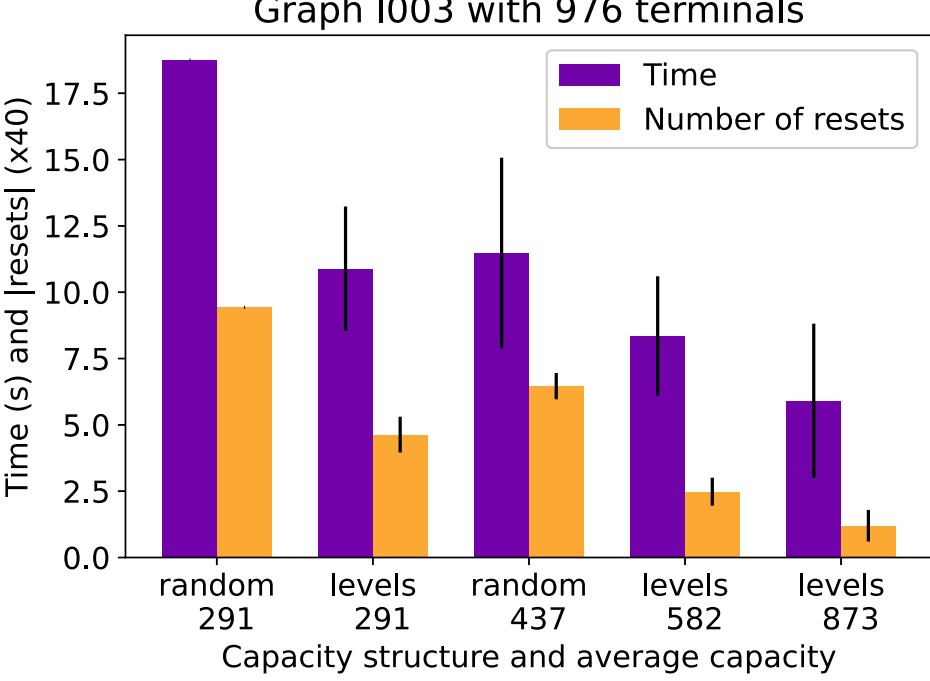

**Fig 11. Vienna I003 time compared by capacity.** The time (purple bars) and number of resets (yellow bars) for 5 different capacity settings.

where the base graph is a road network with a low max degree to develop a heuristic that can quickly find a solution for large instances.

In the literature study, we did not find existing methods capable of solving large CSTPs quickly. While some research has been done on representing graphs quickly using graph convolutional networks, practical algorithms are still in development. This paper demonstrated how an extension of a heuristic for the capacitated STP based on SPH, the Capacitated Shortest Path Heuristic, can find good solutions to the CSTP, while inheriting much of the simplicity of SPH, which allows general input, such as discrete capacities and flexible costs. Despite the generality and the increased problem complexity, CSPH runs in practice in time $O(|V|^2)$ with a small quadratic coefficient (empirically tested). For a sparse graph with 50000 nodes and 1000 terminals, CSPH takes less than 1 minute to find a solution. This is quite acceptable for most practical situations. Furthermore, the running time is strongly influenced by the input properties such as the capacity sizes and the terminal density. Since CSPH is a fast heuristic and the CSPH problem is NP-hard, the heuristic will not guarantee an optimal solution. However, cost estimations on the road network seem to indicate that the optimality gap is acceptable. In conclusion, this paper developed CSPH, a fast cost estimator and initial construction technique for CSTP in sparse graphs. To find a near-optimal solution, the authors recommend using additional methods like local search improvements.

## Author Contributions

**Conceptualization:** Pieter Audenaert.

**Formal analysis:** Simon Van den Eynde.

**Funding acquisition:** Pieter Audenaert, Mario Pickavet.

**Investigation:** Simon Van den Eynde.

**Methodology:** Simon Van den Eynde.

**Project administration:** Pieter Audenaert.

**Resources:** Simon Van den Eynde.

**Software:** Simon Van den Eynde.

**Supervision:** Pieter Audenaert, Mario Pickavet.

**Validation:** Simon Van den Eynde.

**Visualization:** Simon Van den Eynde.

**Writing – original draft:** Simon Van den Eynde.

**Writing – review & editing:** Simon Van den Eynde, Pieter Audenaert, Didier Colle, Mario Pickavet.

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
