## [Decision Letter · Decision Letter 0]

14 Feb 2022

PONE-D-21-39968A construction heuristic for the capacitated Steiner tree problemPLOS ONE

Dear Dr. Van den Eynde,

Thank you for submitting your manuscript to PLOS ONE. After careful consideration, we feel that it has merit but does not fully meet PLOS ONE’s publication criteria as it currently stands. Therefore, we invite you to submit a revised version of the manuscript that addresses the points raised during the review process.

Please read carefully the respected reviewers comments and address them all.

We look forward to receiving your revised manuscript.

Kind regards,

Khalil Abdelrazek Khalil, Ph.D.

Academic Editor

PLOS ONE

Journal Requirements:

3. Please state in your Methods section whether the authors had access to any identifying information on individuals from the telecom dataset, and if so, whether any IRB oversight was in place. If the dataset was anonymised upon receipt, please state this.

Additional Editor Comments:

Please do your best to address all comments raised by respected reviewers and resubmit a revised version.

Reviewers' comments:

Reviewer's Responses to Questions

**Comments to the Author**

1. Is the manuscript technically sound, and do the data support the conclusions?

Reviewer #1: Partly

Reviewer #2: Yes

Reviewer #3: No

2. Has the statistical analysis been performed appropriately and rigorously? 

Reviewer #1: No

Reviewer #2: Yes

Reviewer #3: No

3. Have the authors made all data underlying the findings in their manuscript fully available?

Reviewer #1: Yes

Reviewer #2: Yes

Reviewer #3: Yes

4. Is the manuscript presented in an intelligible fashion and written in standard English?

Reviewer #1: Yes

Reviewer #2: Yes

Reviewer #3: Yes

5. Review Comments to the Author

Reviewer #1: In this paper, the authors work on the heuristic functions for Steiner tree problems. The authors compared and analyzed three heuristic methods, namely linear regression, neural networks and shortest path heuristic (SPH), and suggested that SPH has the best performance. Then a capacitated shortest path heuristic (CSPH) for solving the capacitated Steiner tree problems (CSTP) was developed and analyzed. The paper reads interesting, while my concerns are as follows:

1. The contribution of the paper is not highlighted.

2. The authors spent quite a lot of pages comparing the three heuristics. However, the performance of LR and NN is so poor that the review can hardly be persuaded that they are ideal candidate heuristics.

3. It is intuitive that the searching-based method (SPH) is more accurate but much slower than regression methods. The author could have considered using more powerful regression models, e.g., GBRT, or deep learning models that are more powerful on graphs, e.g., graph convolution networks as the authors mentioned in line 505.

4. In sections 4 and 5, MAPE was used to evaluate the performance of the candidate heuristics. Why not directly use MSE which was used as loss in model training?

5. The method used in feature selection is not persuasive, nor scalable. There are many feature engineering methods in machine learning. However, filtering out zero-valued coefficients in LR model is not robust, as it leads to different results on different datasets, or even might also on different sample sizes.

6. Figures 5 and 6 are wrongly indexed.

7. The authors haven’t discussed/shown whether the CSPH works or how it works. The CSPH is neither compared with ‘ground truth’, nor applied in a solver to evaluate the optimality gap.

8. In line 324, does ‘preprocessing time’ refer to training time?

9. The reviewer doesn’t agree with the statement on line 335. Compared with the difference between SPH and LR/NN, the differences in SPH’s performance on different datasets are neglectable.

10. In table 4, CSPH is compared with SPH and MST. Is SPH applicable to the capacitated problem? If it doesn’t, what’s the point of comparing them, or if it does what’s the point of developing a CSPH?

11. Section 9 shall be extended and provide a more comprehensive analysis, since time performance is the major point in this research.

12. The statement in line 464 about the trend shall be supported by an illustration. Because readers can hardly tell the trend of the numbers in a table.

13. Fig 6 tends to compare the computational time for different given capacity and terminal ratios. However, the graph order, which is a key driver of computational time, is not fixed. This makes the result unreliable. Also, note that a heat map may not be a good choice in this illustration and the authors shall consider other graph types.

14. In line 487 the figure number is wrongly indexed.

15. In line 512

16. A thorough grammar check would be helpful to the paper.

Reviewer #2: According to the minimum STP theory, the authors point out the limitations of INTEGER Linear programs, add constraints, and establish CSPH. Through experimental deduction and analysis, some valuable results are obtained, and the results are verified, including precision, running time and cost. At the same time, the author clearly explains the limitations of the current research conclusions and points out some directions for further research. The experimental data and results in this paper are clear and reasonable, and the results of other researchers are respected in the background description, no dual publication，but there are still some modifications to be made, as follows.

1.This draft does not summarize the contributions of this work. Kindly assist in elucidating the contributions and novelties.

2.The introduction is clearly explained the significant of the study, and relevant literature as well as the proposed method. Nevertheless, please state the study objectives.

3.In the section 'introduction', the relative studies were not logically reviewed in both time and dimension. The section of introduction should be restructured in my opinion. This section introducing the reader to the existing literature. While doing this, they introduce authors and their areas of study in general terms without presenting their specific findings. But in this manuscript, such descriptions were not existed.

4.Without explanation, abbreviations such as 'MSE' and 'DFS' are introduced. Before any abbreviations are established, the complete terminology should be provided. Additionally, it would be beneficial for the authors to provide a list of abbreviations prior to the technical session.

Reviewer #3: Capacitated Steiner tree problem is one of classical combinatorial optimization problems. This paper presents a heuristic approach to calculate capacitated Steiner tree which proves to be fast and effective on sparse graph.

However, the proposed algorithm efficiency is proved by limited experimental test based on graphical dataset, which lacks basic algorithm complexity analysis and rigorous proof. What's more the optimality comparison among the proposed Heuristic Technique, Linear Regression Model and a Feedforward Fully Connected Deep NN is lack of persuasion. It is partly because the result could be influenced by latent factors such as the programmers' coding technique.

6. PLOS authors have the option to publish the peer review history of their article (what does this mean?). If published, this will include your full peer review and any attached files.

Reviewer #1: No

Reviewer #2: No

Reviewer #3: No

---

## [Author Response · Author response to Decision Letter 0]

11 Apr 2022

[Use a markdown viewer for an improved reading experience]

# Response to Reviewers

## Academic editor

1. Attention was paid to file naming - this should be ok now.

2. The code is available at https://github.com/UGent-DNA/CSPH

3. The Comsof telecom dataset was removed from the experiments. The Vienna telecom dataset was properly anonymised, see https://homepage.univie.ac.at/ivana.ljubic/research/STP/.

4. All the datasets used in the experiments are publicly available at the following repositories:

 * PUC dataset: http://steinlib.zib.de/showset.php?PUC

 * LIN dataset: http://steinlib.zib.de/showset.php?LIN

 * Vienna dataset: https://homepage.univie.ac.at/ivana.ljubic/research/STP/

## Reviewer 1

1. The contribution of the paper is not highlighted.

 * The abstract, introduction, and conclusion were expanded to clarify the contribution.

2. *The authors spent quite a lot of pages comparing the three heuristics. However, the performance of LR and NN is so poor that the review can hardly be persuaded that they are ideal candidate heuristics.*

 * The analysis on LR and NN was removed. Instead, an extension of the literature study shows that the current state-of-the-art neural networks are not yet capable of solving large-scale STPs and by extension CSTPs, see section 2.2. 

3. *It is intuitive that the searching-based method (SPH) is more accurate but much slower than regression methods. The author could have considered using more powerful regression models, e.g., GBRT, or deep learning models that are more powerful on graphs, e.g., graph convolution networks as the authors mentioned in line 505.*

 * See response point 2. 

4. *In sections 4 and 5, MAPE was used to evaluate the performance of the candidate heuristics. Why not directly use MSE which was used as loss in model training?*

 * No longer relevant, see response point 2.

5. *The method used in feature selection is not persuasive, nor scalable. There are many feature engineering methods in machine learning. However, filtering out zero-valued coefficients in LR model is not robust, as it leads to different results on different datasets, or even might also on different sample sizes.*

 * See response point 2.

6. *Figures 5 and 6 are wrongly indexed.*

 * The figures have been remade and should be indexed correctly.

7. *The authors haven’t discussed/shown whether the CSPH works or how it works. The CSPH is neither compared with ‘ground truth’, nor applied in a solver to evaluate the optimality gap.*

 * In section 4, it is explained how CSPH works. For an implementation of the code, see https://github.com/UGent-DNA/CSPH.

 * It was not possible within the scope of the review to apply a solver and evaluate the optimality gap.

 * A rough comparison was made by comparing with Dijkstra (which gives optimal fiber cost) and SPH gives an estimation (on the tested graphs, should be within $5\\%$ of optimal) of where the optimal trenching cost is since SPH does not take into account capacities. See section 7.1.

 * This does not give a concrete optimality gap but shows that CSPH returns "decent" solutions that are not arbitrarily bad.

8. *In line 324, does ‘preprocessing time’ refer to training time?*

 * No longer relevant. See response point 2.

9. *The reviewer doesn’t agree with the statement on line 335. Compared with the difference between SPH and LR/NN, the differences in SPH’s performance on different datasets are neglectable.*

 * No longer relevant. See response point 2.

10. *In table 4, CSPH is compared with SPH and MST. Is SPH applicable to the capacitated problem? If it doesn’t, what’s the point of comparing them, or if it does what’s the point of developing a CSPH?*

 * SPH is not applicable to the capacitated problem directly, but it does provide an estimation of an optimal solution if no capacities would be present. In fact, on a graph with no capacities, CSPH should behave the same as SPH. But more importantly, it gives a benchmark for time comparison, so that it can be realistically estimated how long CSTP will take based on SPH time measurements on a different system.

11. *Section 9 shall be extended and provide a more comprehensive analysis, since time performance is the major point in this research.*

 * Section 9, reworked to section 7.2 was expanded. A theoretical worst-case analysis was added. The experiments were redone. 

12. *The statement in line 464 about the trend shall be supported by an illustration. Because readers can hardly tell the trend of the numbers in a table.*

 * Figs 5-7 now indicate the evolution over time when graph size increases for CSPH. These figures also contain a linear and quadratic function to facilitate the estimation of a trend from the figure.

13. *Fig 6 tends to compare the computational time for different given capacity and terminal ratios. However, the graph order, which is a key driver of computational time, is not fixed. This makes the result unreliable. Also, note that a heat map may not be a good choice in this illustration and the authors shall consider other graph types.*

 * The heat map was removed and replaced by different figures

 * Figs 8-9 now enable the estimation of the effect of terminal ratios by comparing the gaps between the lines for a fixed number of nodes

 * Figs 10-11 show the effect of capacity settings on a single graph, thus fixing the graph order.

14. *In line 487 the figure number is wrongly indexed.*

 * This should be fixed

15. *In line 512*

16. *A thorough grammar check would be helpful to the paper.*

 * The paper was reread multiple times and should have an improved grammatical structure

## Reviewer 2

1. *This draft does not summarize the contributions of this work. Kindly assist in elucidating the contributions and novelties.*

 * The abstract, introduction, and conclusion were expanded to clarify the contribution.

2. *The introduction is clearly explained the significant of the study, and relevant literature as well as the proposed method. Nevertheless, please state the study objectives.*

 * The study objectives are clarified in the introduction.

3. *In the section 'introduction', the relative studies were not logically reviewed in both time and dimension. The section of introduction should be restructured in my opinion. This section introducing the reader to the existing literature. While doing this, they introduce authors and their areas of study in general terms without presenting their specific findings. But in this manuscript, such descriptions were not existed.*

 * Ordering the studies chronologically, while still maintaining a clear separation between research on STP, neural networks and CSTP proved difficult. Therefore, it was decided to order the papers chronologically within each subsection, as much as possible.

 * The literature study was expanded by adding more information on the cited papers, in particular, adding information on the techniques used and the results achieved in the cited papers. 

4. *Without explanation, abbreviations such as 'MSE' and 'DFS' are introduced. Before any abbreviations are established, the complete terminology should be provided. Additionally, it would be beneficial for the authors to provide a list of abbreviations prior to the technical session.*

 * No list of abbreviations was added, but all abbreviations were re-evaluated. Unnecessary abbreviations were removed and we payed extra attention to make sure all abbreviations were properly explained on first use.

## Reviewer 3

*However, the proposed algorithm efficiency is proved by limited experimental test based on graphical dataset, which lacks basic algorithm complexity analysis and rigorous proof. What's more the optimality comparison among the proposed Heuristic Technique, Linear Regression Model and a Feedforward Fully Connected Deep NN is lack of persuasion. It is partly because the result could be influenced by latent factors such as the programmers' coding technique.*

* The time experiments were changed to make it easier to evaluate the figures and draw conclusions.

* A theoretical worst-case time analysis was written, see section 7.2.

* The analysis on LR and NN was removed. Instead, an extension of the literature study shows that the current state-of-the-art neural networks are not yet capable of solving large-scale STPs and by extension CSTPs, see section 2.2.

---

## [Decision Letter · Decision Letter 1]

6 Jun 2022

A Construction Heuristic for the Capacitated Steiner Tree Problem

PONE-D-21-39968R1

Dear Dr. Van den Eynde,

We’re pleased to inform you that your manuscript has been judged scientifically suitable for publication and will be formally accepted for publication once it meets all outstanding technical requirements.

Kind regards,

Khalil Abdelrazek Khalil, Ph.D.

Academic Editor

PLOS ONE

Additional Editor Comments (optional):

Thank you for submitting a revised version of your manuscript.

Reviewers' comments:

Reviewer's Responses to Questions

**Comments to the Author**

1. If the authors have adequately addressed your comments raised in a previous round of review and you feel that this manuscript is now acceptable for publication, you may indicate that here to bypass the “Comments to the Author” section, enter your conflict of interest statement in the “Confidential to Editor” section, and submit your "Accept" recommendation.

Reviewer #1: All comments have been addressed

Reviewer #2: All comments have been addressed

2. Is the manuscript technically sound, and do the data support the conclusions?

Reviewer #1: (No Response)

Reviewer #2: (No Response)

3. Has the statistical analysis been performed appropriately and rigorously? 

Reviewer #1: (No Response)

Reviewer #2: (No Response)

4. Have the authors made all data underlying the findings in their manuscript fully available?

Reviewer #1: (No Response)

Reviewer #2: (No Response)

5. Is the manuscript presented in an intelligible fashion and written in standard English?

Reviewer #1: (No Response)

Reviewer #2: (No Response)

6. Review Comments to the Author

Reviewer #1: (No Response)

Reviewer #2: (No Response)

7. PLOS authors have the option to publish the peer review history of their article (what does this mean?). If published, this will include your full peer review and any attached files.

Reviewer #1: No

Reviewer #2: No
